

# High-precision $\delta^{13}$C-CO$_2$ analysis from 1 mL of ambient atmospheric air via continuous flow IRMS: from sampling to storage to analysis.

Joana Sauze[1], Marie-Laure Tiouchichine[1], Alexandru Milcu[1,2], Clément Piel[1]

[1]Ecotron Européen de Montpellier (UAR 3248), Univ Montpellier, CNRS, Montferrier-sur-Lez, France

[2]CEFE, Univ Montpellier, CNRS, EPHE, IRD, 34293, Montpellier, France

Corresponding author: Joana Sauze (joana.sauze@cnrs.fr)

**Abstract.** The carbon isotopic composition ($\delta^{13}$C) of atmospheric carbon dioxide (CO$_2$) is a key tracer for understanding terrestrial carbon dynamics, yet its application in small-volume sampling systems remains constrained by analytical limitations. Here, we present a novel methodology for high-precision $\delta^{13}$C analysis of ambient atmospheric CO$_2$ from 1 mL air samples, tailored to the challenges of growth chamber experiments using microcosm model systems and other volume-limited systems. Our approach emerged from testing the effects of custom vial conditioning, dual-sealing with Terostat®, ultra-low-temperature storage at -80°C, and cryogenic pre-concentration coupled to continuous-flow isotope-ratio mass spectrometry (IRMS). We demonstrate that vial conditioning and improved dual sealing are critical to ensure analytical precision. Our combined method achieves a precision of ± 0.1 ‰ on $\delta^{13}$C measurements, with negligible isotopic drift for storage durations up to 1-week if ultra-low-temperature storage and zip-lock bags full of CO$_2$-free air were used. Longer storage times reduces measurement precision, emphasising the importance of short-term preservation. This technique offers a significant advance for carbon stable isotope applications in constrained environments, enabling minimally invasive, high-frequency $\delta^{13}$C monitoring with good precision at the millilitre scale.

## 1 Introduction

The stable carbon isotopic composition of atmospheric CO$_2$ ($\delta^{13}$C) is a powerful tool for tracing carbon sources and sinks, quantifying biogeochemical processes, and constraining global carbon cycle models (Bowling et al., 2008; Ciais et al., 2014; Farquhar et al., 1989). Because key processes such as photosynthesis and respiration fractionate carbon isotopes differently, isotopic measurements provide insights into the balance and dynamics of terrestrial carbon fluxes. These isotopic differences are essential for constraining global carbon cycle models and improving predictions of carbon–climate feedbacks (Tans et al., 1993).

Recent advances in isotope ratio mass spectrometry (IRMS) and automated gas handling systems have considerably improved the efficiency and accessibility of $\delta^{13}$C analysis. Continuous-flow techniques and the development of more stable and sensitive detectors now allow for the rapid processing of large numbers of samples at reduced cost and with high analytical precision





(Brand, 1996; Fisher et al., 2006; West et al., 2006). These improvements have made stable isotope analysis more widely applicable in ecosystem and atmospheric sciences. However, most of these methods still require sample volumes of several millilitres (Tu et al., 2001), limiting their use in situations where only small amounts of air can be collected, or where repeated
sampling is necessary. This volume requirement remains a key bottleneck in experimental systems where space, atmosphere volume, or sample handling are constrained.

Laser-based instruments, such as those developed by Picarro analysers (Picarro Inc., Santa Clara, USA), have emerged as promising alternatives for in situ $\delta^{13}C$ analysis. However, these systems typically require sample volumes of around 12 mL to achieve sufficient sensitivity as well as high precision. Simple dilution of smaller samples (e.g., 1 mL into 12 mL) is not
feasible, as it reduces $CO_2$ concentrations below the detection limits of these instruments. Thus, despite their operational advantages, laser-based methods are currently not suited for high-precision analysis of sub-1 mL atmospheric air sample.

These constraints are particularly evident in growth chamber experiments, where plants are cultivated in controlled environments under tightly regulated atmospheric conditions. One widely used method to disentangle $CO_2$ sources is the Keeling plot approach, which estimates the isotopic signature of ecosystem respiration by examining the linear relationship
between $\delta^{13}C$ and the inverse of $CO_2$ concentration during periods of atmospheric mixing (Keeling, 1958; Pataki et al., 2003). The strength of such analyses lies in the precision and temporal resolution of $\delta^{13}C$ measurements, which depend on the ability to capture subtle isotopic variations in the atmosphere, often under constraints of sampling frequency, volume, and system disturbance (Midwood and Millard, 2011; Pataki et al., 2003; Sperlich et al., 2022; Werner et al., 2006). Such setups often involve a large number of small pots or microcosms, with limited headspace available for gas sampling (Gillespie et al., 2020;
Guillot et al., 2019; Siegwart et al., 2023). In these systems, withdrawing large volumes of air can disrupt the experimental conditions or interfere with repeated measurements. As a result, there is a growing need for analytical methods capable of measuring $\delta^{13}C$ in very small air volumes (~1 mL) at ambient $CO_2$ mixing ratio, while maintaining high analytical precision. Such methods would allow for high-resolution, minimally invasive sampling across space and time, and facilitate isotopic monitoring in highly replicated experimental designs.

To this end, we developed a novel analytical workflow for measuring $\delta^{13}C$ of ambient atmospheric $CO_2$ in 1 mL samples, based on continuous-flow isotope ratio mass spectrometry. Our approach builds upon earlier cryogenic trapping configurations originally developed for small carbonate samples (Fiebig et al., 2005), which we adapted and optimised for ambient $CO_2$ mixing ratio. This configuration significantly reduces the required sample volume while maintaining high analytical performance, with a precision of $\pm$ 0.1 ‰ on $\delta^{13}C$, opening the door to new experimental designs in volume-constrained
experimental systems.

In addition to challenges associated with small atmospheric sample volumes, the storage of gas samples prior to analysis presents a major limitation. Isotopic composition can drift due to preferential diffusion of lighter isotopes, leakage, or physicochemical interactions with storage materials, especially over time. This issue has been documented for various isotopes, including $^{13}C$, $^{18}O$, $^{15}N$, and $^2H$ (Hardie et al., 2010; Kuehfuss et al., 2014; Laughlin and Stevens, 2003; Mortazavi and Chanton,



2002; Nauer et al., 2021; Nelson, 2000; Paul and Skrzypek, 2006). Current protocols thus require analysis within hours of sampling, restricting laboratory collaboration and field sampling efforts.

As part of our methodological development, we designed and specifically tested the impacts of: 1) pre-conditioning vials by flushing with $CO_2$-free air, 2) simple vs. double septum configurations, 3) the application of an additional Terostat® layer at the bottom of the cap, 4) storage temperature and 5) storage duration. These tests led to the development of a sample

preservation strategy that extends the storage time of small air samples without compromising isotopic integrity, thereby improving both the flexibility and robustness of $\delta^{13}C$ measurements from small atmospheric samples.

In this paper, we present this integrated methodology—from sampling to storage to analysis—and demonstrate its application in the context of controlled-chamber experiments as well as its compatibility for field experiments. We assess its precision, and suitability for high-frequency isotopic monitoring, and discuss its broader potential for advancing carbon cycle research.

Our method meets the targeted precision of ± 0.1 ‰ on $\delta^{13}C$ using only 1 mL of ambient atmospheric air, thereby offering a powerful tool especially for studying carbon dynamics in highly constrained experimental settings.

## 2 Material and methods

### 2.1 Vial conditioning: the basics

We used 5.9 mL flat bottom soda exetainers (Labco Limited, UK) with single chlorobutyl septum as sampling vials for $CO_2$

analysis. After closing, the vials were conditioned on a custom-built manifold designed to prepare them for trace-level $CO_2$ sampling (Fig. 1). This manifold (Fig. 1) is constructed using Swagelok® fittings and valves (Swagelok Company, USA). It consists of twelve interconnected ¼-inch tees (SS-400-3), linked to two manual valves: a two-way valve (SS-43GS4) for vacuum control and a three-way valve (SS-43GXS4) for pressurised nitrogen supply and pressure equilibration. Luer needles (27G ¾", Terumo Agani, China) were inserted into Ultra-Torr® fittings (SS-4-UT-6-400), which were connected to each tee.

Batches of 12 vials were evacuated simultaneously to a pressure of $10^{-1}$ bar over 8 minutes, then filled with nitrogen gas (Alphagaz 1, Air Liquid, France) at 0.5 bars for 20 seconds to establish an overpressure. Nitrogen was used instead of synthetic air in order to minimize nitrous oxides production in the IRMS source, and therefore potential bias with $CO_2$ measurements. By contrast, helium was not used in order to minimize diffusion through the septum during storage (data not shown). Finally, to restore atmospheric pressure, the over pressurised vials were vented by opening the exhaust valve for 5 seconds. This

evacuation-filling cycle, which included a final equilibration to ambient atmospheric pressure, was repeated four times to ensure thorough removal of any residual $CO_2$ or contaminants and to create a reproducible, $CO_2$-free starting atmosphere. When the four evacuation-filling cycles were completed, the processed vials were equilibrated to ambient atmospheric pressure, not under vacuum, and ready for gas sampling. A control set vials (i.e. blanks) was analysed and showed no detectable $CO_2$ signal, confirming the integrity of the conditioning protocol.




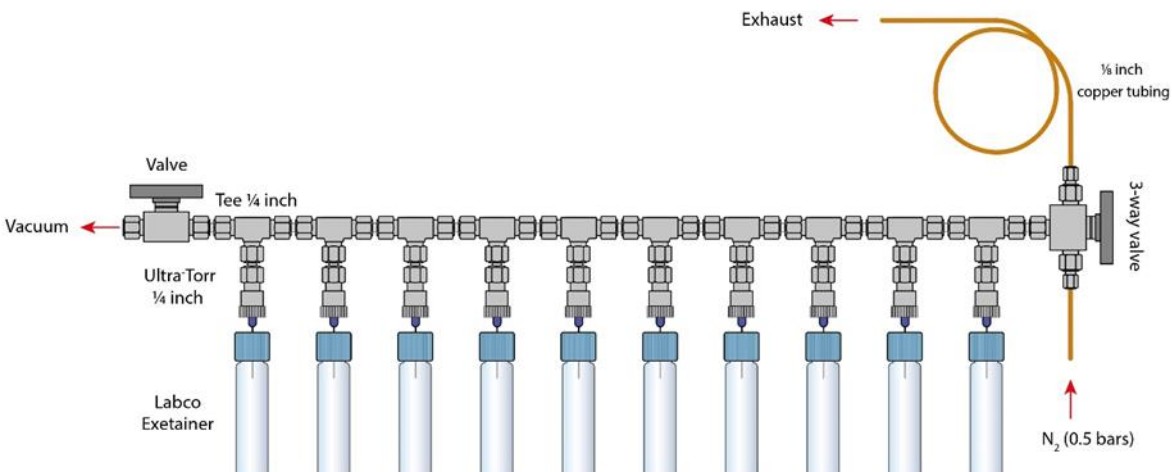

**Figure 1: Schematic of the vial conditioning manifold. For clarity only 10 vials positions are shown. The conditioning procedure includes: (1) vacuum of the vials to $10^{-1}$ bar (~8 minutes), (2) filling with $N_2$ gas at 0.5 bars with overpressure (~20 seconds), and (3) restoration to atmospheric pressure (~5 seconds). A 1/8-inch copper tube is connected to the exhaust outlet to prevent ambient $CO_2$ from diffusing into the vials during pressure equilibration. Steps 1, 2 and 3 were repeated four times.**

## 2.2 Gas sampling

Gas sampling was done using a 1 mL syringe (Soft-Ject, Germany) equipped with a Luer needle 27G ¾ (Terumo Agani, China). The syringe was first flushed three times with ambient air to remove any residual gases. It was then flushed 3 times with the targeted gas, by injecting and withdrawing it inside the container (chamber, flask or jar for example) without sampling or releasing, to avoid any contamination from prior samples. The syringe was then filled with a volume of air slightly exceeding the targeted volume (1 mL) before being withdrawn from the container. The volume in the syringe was carefully adjusted to 1 mL by expelling the excess air. The air sample was subsequently injected into a simple septum vial through the septum. After injection, Terostat® was reapplied over the septum, to prevent any gas leakage.

## 2.3 Isotopes analysis

Isotopic analyses of $CO_2$ were performed using a continuous-flow isotope ratio mass spectrometer (IRMS; Delta V Plus, Thermo Fisher Scientific, USA) coupled to a GasBench II preparation system and a ConFlo IV. The analytical setup included an automated cryofocus unit (cryogenic trapping), initially designed for carbonate analysis (Fiebig et al., 2005), but adapted here for high-precision $\delta^{13}C$ measurements of trace-level $CO_2$ in small air samples.

The PAL autosampler initiated each run by moving the needle to the appropriate exetainer, which was then continuously flushed with helium, using a single flushing needle that is fixed to the PAL, to eliminate atmospheric gases. The automated protocol (Table 1) then began: the air sample was introduced into the GasBench II via an automated injection system, where water vapor was removed using a Nafion® membrane. Ten seconds after flushing, the cryofocus unit was lowered into liquid



nitrogen, and 20 s later, the Valco valve switched to "load" mode for 360 s to allow $CO_2$ to condense while non-condensable gases were flushed by the helium stream (Brand et al., 2010).

Simultaneously, five rectangular reference $CO_2$ peaks were injected through the open split by the ConFlo IV, with $\delta^{13}C$ values reported relative to the Vienna Pee Dee Belemnite (VPDB) scale. The fifth peak served as the reference for sample calibration. At 390 s, the Valco valve returned to "inject" mode, and after a 20 s delay, the cryofocus was raised, releasing the trapped $CO_2$ into the helium carrier stream via sublimation. The gas then passed through a second Nafion® trap to remove residual water, followed by a gas chromatography column (PoraPlot Q, 25 m, 0.32 mm, 10 µm film, Agilent) held at 35°C, which enabled

separation of $CO_2$ from gases such as $O_2$ and $N_2$, thereby ensuring stable ionisation conditions in the mass spectrometer. $CO_2$ was finally introduced into the Delta V Plus via an open split in the ConFlo IV. The analysis also included a $CO_2$ blanking procedure to correct for any background signal originating from the analytical system itself (Paul et al., 2007).

In contrast to traditional injection systems (Spötl, 2004; Spötl and Vennemann, 2003), this configuration produces a single, well-defined $CO_2$ peak, as the entire analyte is injected in the ion source in once. The $\delta^{13}C$ values were reported relative to the

Vienna Pee Dee Belemnite (VPDB) standard and calibrated using a working $CO_2$ standard that had been referenced against a referenced cylinder. Three vials containing 1 mL of $CO_2$ working standard ($CO_2$ in synthetic air, Air Liquide, France) were analysed at the beginning and the end of each daily run to correct for instrumental drift and ensure analytical precision. Analysis time per sample was 17 minutes, and analytical precision consistently reached ± 0.1 ‰ for $\delta^{13}C$.

**Table 1: Isodat software-based protocol for isotopic analysis of $\delta^{13}C$ using a Delta V plus IRMS coupled to a Gas Bench II, a ConFlo IV as well as an automated cryofocus unit. The table summarises the timing and sequence of key operations controlled by the PAL autosampler and GasBench II system, including activation of the cryofocus trap (Trap), switching of the Valco valve (Valco), and triggering of the $CO_2$ blanking function ($CO_2$ blanking). Reference automatic represents the $CO_2$ standards gas capillary that is repeatedly activated (On) and deactivated (Off) to generate rectangularly shaped signals of mass 44.**

| Time (s) | Reference automatic | Valco | Trap | $CO_2$ blanking |
|----------|---------------------|-------|------|-----------------|
| 2 | | Inject | | Off |
| 10 | On | | Down | |
| 30 | Off | Load | | |
| 50 | On | | | |
| 70 | Off | | | |
| 85 | | | | Off |
| 240 | On | | | |
| 260 | Off | | | |
| 280 | On | | | |
| 300 | Off | | | |





| 380 | | | On |
|-----|-----|-----|-----|
| 390 | | Inject | |
| 400 | On | | |
| 410 | | Up | |
| 420 | Off | | |
| 440 | | | Off |
| 770 | | | On |

**2.4 Statistical analyses**

All statistical analyses and visualisations were performed using R software (R-4.4.2, R Core Team, 2015). To assess significant differences in $\delta^{13}C$ values between two sample groups, Welch's t-tests were applied when variances were unequal, while Student's t-tests were used when the assumption of equal variances held. For datasets involving more than two groups with unequal variances, a Welch's ANOVA was performed, followed by a Games–Howell post-hoc test to evaluate pairwise

differences. Standard deviations were calculated for each group to evaluate the precision of the measurements.

**3 Results**

**3.1 Pre-conditioning flush with $CO_2$-free air on vial**

To achieve high-precision $\delta^{13}C$ measurements from small ambient atmospheric sample, we identified the initial vial flush step as a critical factor. To minimize background contamination and prevent isotopic fractionation due to degassing, each vial was

systematically flushed for 8 seconds with dry $CO_2$-free air prior to be conditioned (section 2.1). This pre-conditioning step was performed manually by briefly removing and reinserting the septum cap while applying the continuous $CO_2$-free air flow into the vial, in order to remove residual $CO_2$ and stabilise internal pressure, thereby creating optimal conditions for repeatable measurements for the subsequent conditioning steps.

To assess the actual impact of this flushing step, we conducted a comparative experiment using vials prepared either with or

without this $CO_2$-free air flush. In both cases, 1 mL of a $CO_2$ working standard ($\delta^{13}C$ = -38.72 ‰, $CO_2$ in synthetic air, Air Liquide, France) was introduced after conditioning, and samples were immediately analysed by IRMS. In the absence of the flushing step, the results exhibited poor precision, ± 0.54 ‰. In contrast, when vials were flushed with $CO_2$-free air before conditioning, precision improved, with standard deviations reaching ± 0.09 ‰ (Fig. 2).

A Welch's t-test (assuming unequal variances) confirmed that the difference between the means of the two groups was

statistically significant (p < 0.05), highlighting the critical role of the flushing step in ensuring high precision of isotopic measurements.



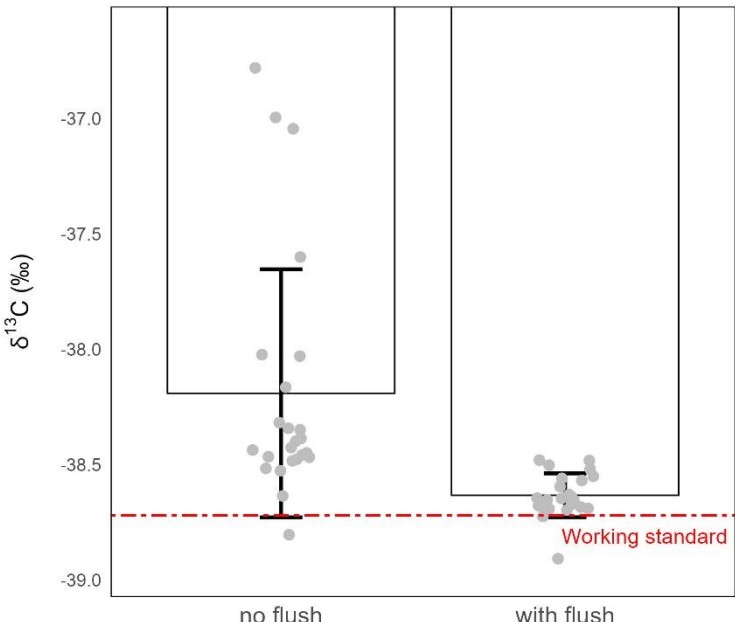

**Figure 2: Effect of the initial $CO_2$-free air flush on $\delta^{13}C$ analysis in 1 mL gas samples. $\delta^{13}C$ values (‰ vs VPDB) measured from vials**
**conditioned either without a flushing step (left) or with an 8-second $CO_2$-free air flush (right) prior to gas introduction. The red dashed line indicates the expected isotopic value of the working standard (-38.72 ‰). Grey circles represent individual replicate measurements. Bars show the mean ± standard deviation for each condition. Without flushing, the measured precision was ± 0.54 ‰ (n = 24), whereas with flushing, precision was ± 0.09 ‰ (n = 23).**

### 3.2 Analytical performance of 1 mL ambient atmospheric air samples

A typical chromatogram produced by the method is described in Fig. 3. This chromatogram exhibits a single well-defined $CO_2$ peak with sharp Gaussian symmetry and excellent signal-to-noise ratio. The retention time remains stable across replicate injections, indicating consistent flow dynamics and thermal stability of the GC column. Baseline separation is maintained, with no detectable co-elution or interfering species in the target m/z acquisition window. Peak integration was performed within fixed boundaries, and total $CO_2$ signal area was sufficient to ensure accurate $\delta^{13}C$ calculation despite the low analyte
mass (amplitude from 2 to 2.5 V). Across a series of replicate measurements (n = 24), performed on independently prepared vials (no storage, $CO_2$-free air flush), the analytical precision was consistently within ± 0.10 ‰.





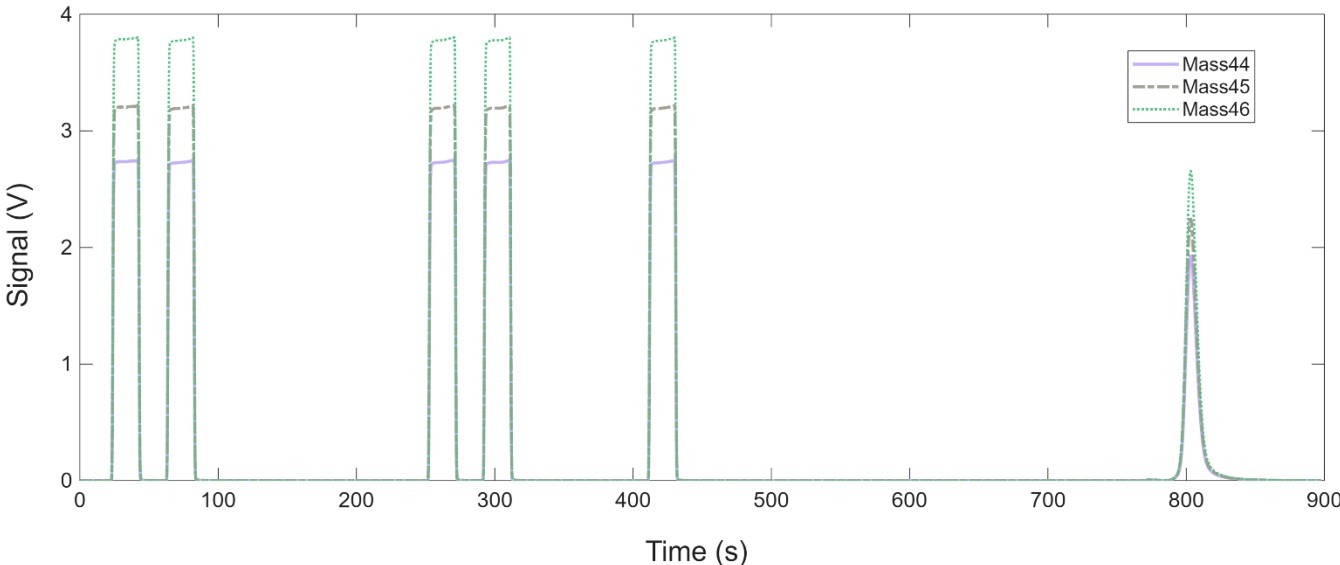

**Figure 3: Representative chromatogram obtained from the analysis of 1 mL of atmospheric air using an IRMS equipped with**
**GasBench II and a ConFlo IV. The ion intensities for m/z 44 (purple solid line), 45 (brown dot dash line), and 46 (green dotted line),**
**corresponding to the different isotopologues of carbon dioxide. The peaks observed between 0 and 450 seconds correspond to**
**successive injections of a $CO_2$ reference gas, while the single peak around 800 seconds corresponds to the atmospheric air sample.**

In situations where $CO_2$ concentrations exceed atmospheric levels—such as in soil incubations where soil respiration leads to
accumulation of $CO_2$ in the air space, experiments involving [13]C-labelled glucose additions or microbial incubations—the
same analytical protocol can be applied to sample volumes as low as 50 µL without compromising precision (Siegwart et al.,
2023). The enhanced signal associated with elevated $CO_2$ concentrations ensures that even these small volumes generate
chromatograms with sufficient peak intensity and resolution for accurate isotopic analysis.

**3.3 Effect of septum configuration on short-term storage stability**

To assess the impact of septum configuration on the preservation of isotopic integrity during short-term storage, we compared
vials sealed with a single septum to those fitted with a double septum (Labco Limited, UK). In both cases, 1 mL of working
standard ($\delta^{13}C$ = -38.72 ‰, $CO_2$ in synthetic air, Air Liquide, France) was injected into the vials, which were then stored for
24 hours at room temperature. Isotopic analyses revealed no statistically significant difference between the two configurations
(t-test, p > 0.05, assuming equal variances; Fig. 4). However, in both cases, the precision exceeded 0.1 ‰ (± 0.20 ‰ for simple
195 septum vs ± 0.16 ‰ for double septum), indicating isotopic drift during storage.



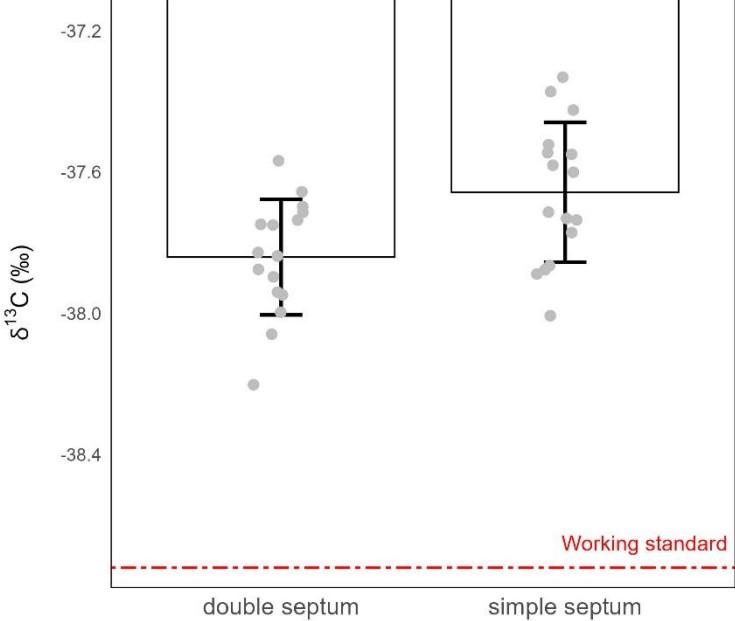

**Figure 4: Effect of septum configuration on $\delta^{13}$C stability during 24-hour storage at ambient temperature. $\delta^{13}$C values (‰ vs VPDB) measured in 1 mL working standard in vials sealed with either a single septum (right) or a double septum (left). The red dashed line represents the expected isotopic value of the working standard (-38.72 ‰). Grey circles represent individual replicate measurements. Bars show the mean ± standard deviation for each condition. Precision was ± 0.16 ‰ for the double septum (n=16) and ± 0.20 ‰ for the simple septum (n=16).**

Although the double septum offered no measurable improvement in performance, it comes at a higher price. For this reason, and in the absence of any significant benefit, the single septum configuration was retained for subsequent analyses. These results suggest that while septum configuration alone does not mitigate storage-related isotopic shifts, alternative sealing or preservation strategies must be explored to improve the long-term stability of small-volume air samples.

### 3.4 Effect of dual-sealing on short-term storage performance

To further improve sample integrity during short-term storage, we evaluated the impact of applying Terostat® not only on the top of the caps but also around the bottom, near the thread area. This dual-sealing technique aims to significantly reduces the risk of gas leakage, diffusion or isotopic drift, particularly over longer storage periods. As in previous tests, 1 mL of working standard ($\delta^{13}$C = -36.26 ‰, $CO_2$ in synthetic air, Air Liquide, France) was injected into each vial, which was then stored for 24 hours at room temperature. The results showed a clear improvement in measurement precision when dual-sealing Terostat® was used (Fig. 5). With simple-sealing, the precision was limited, with a standard deviation of ± 0.22 ‰ while when Terostat® dual-sealing was applied, precision improved to ± 0.11 ‰.





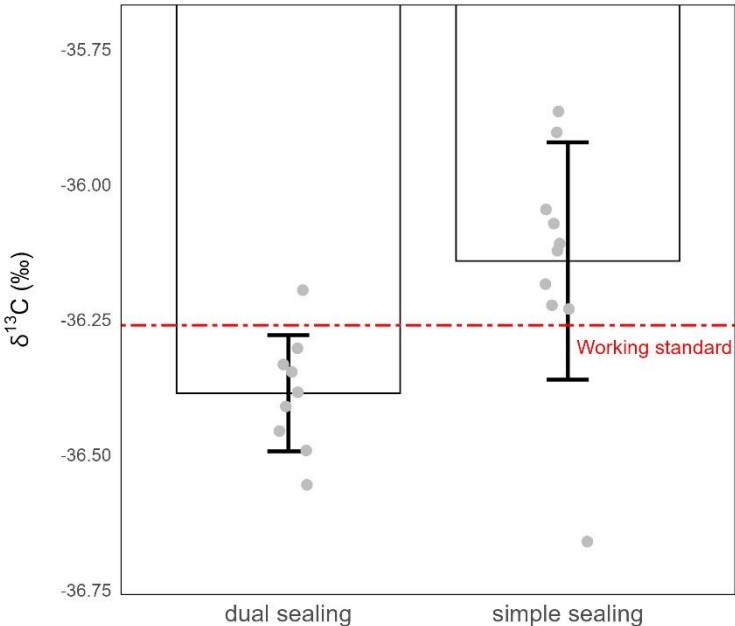

**Figure 5: Effect of sealing improvement on δ¹³C stability during 24-hour storage at ambient temperature. δ¹³C values (‰ vs VPDB) measured in 1 mL working standard in vials with simple- (right) or dual-sealing (left) Terostat® addition on top of the septum. The red dashed line represents the expected isotopic value of the working standard (-36.26 ‰). Grey circles represent individual replicate measurements. Bars show the mean ± standard deviation for each condition. Precision was ± 0.22 ‰ without Terostat® addition (n=10) and ± 0.11 ‰ with dual-sealing (n=9).**

A Welch's t-test (unequal variances) confirmed that the difference in mean $\delta^{13}C$ values between the two conditions was statistically significant ($p < 0.05$). Interestingly, $\delta^{13}C$ values in the two conditions deviated from the working standard in opposite directions, suggesting that the source of contamination was not the same. Nevertheless, although this dual-sealing method brings performance closer to the desired analytical precision, the target threshold of ± 0.1 ‰ was not reached, indicating that additional measures—such as temperature control or more robust barrier systems—may be necessary to ensure optimal storage stability.

**3.5 Influence of storage temperature on isotopic signal stability of small-volume air**

To assess whether the target precision of ± 0.1 ‰ for $\delta^{13}C$ measurements could be achieved on 1 mL air stored samples, we investigated the effect of storage temperature on isotopic stability (Kornfeld et al., 2012). Vials were flushed with $CO_2$-free air, sealed using the Terostat® dual-sealing method, 1 mL of sample was injected ($\delta^{13}C$ = -35.20 ‰, $CO_2$ in synthetic air, Air Liquide, France), and then stored for 24 hours under three different conditions: room temperature (~20°C), -20°C, and -80°C. To further minimize the risk of contamination, the vials were placed in sealed zip-lock bags, which were themselves filled with $CO_2$-free air. In the event of contamination, the gas entering the vials is $CO_2$-free air, thus limiting any disturbance to the isotopic signal measured by IRMS.



Isotopic measurements clearly indicated that lower storage temperatures improved measurement precision (Fig. 6). At room temperature and -20 °C, precision remained above the target threshold (± 0.50 ‰ and ± 0.24 ‰ respectively). Only at -80 °C

was the desired precision of ± 0.1 ‰ reliably achieved.

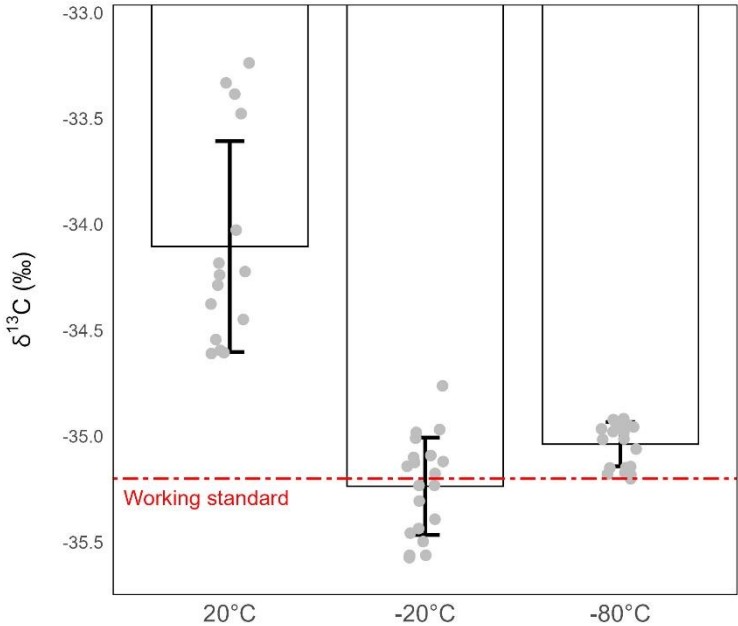

**Figure 6: Effect of storage temperature on δ¹³C stability during 24-hour storage at ambient temperature (left), -20°C (middle), -80°C (right). δ¹³C values (‰ vs VPDB) were measured from 1 mL working standard. The red dashed line represents the expected**
**isotopic value of the working standard (-35.20 ‰). Grey circles represent individual replicate measurements. Bars show the mean ± standard deviation for each condition. Precision was ± 0.50 ‰ for sample stored at room temperature (n=15), ± 0.24 ‰ for sample stored at -20°C (n=20) and ± 0.10 ‰ for sample stored at -80°C (n=20).**

To verify that no contamination or leakage could bias the results at low $CO_2$ levels, a control set of empty vials (flushed, dual-
sealed and conditioned but not filled with $CO_2$) was stored at -80 °C and subsequently analysed. These blank vials showed no detectable $CO_2$ signal, confirming the integrity of the sealing protocol and the absence of background contamination during storage under ultra-low-temperature conditions.

Based on this finding, a key remaining question concerns the maximum duration over which samples can be stored at -80°C without significant isotopic drift.

**3.6 Effect of storage duration at -80°C on isotopic stability**

To determine the maximum duration over which small-volume air samples can be stored at -80°C, without compromising isotopic signal, we analysed vials containing 1 mL of two working standards (δ¹³C = -35.2 ‰ and δ¹³C = -36.26 ‰, $CO_2$ in




synthetic air, Air Liquide, France) stored for 1, 2 and 4 weeks. All samples were flushed with $CO_2$-free air and Terostat® dual-sealing was applied prior to be stored at -80°C in zip-lock bags full of dry $CO_2$-free air.


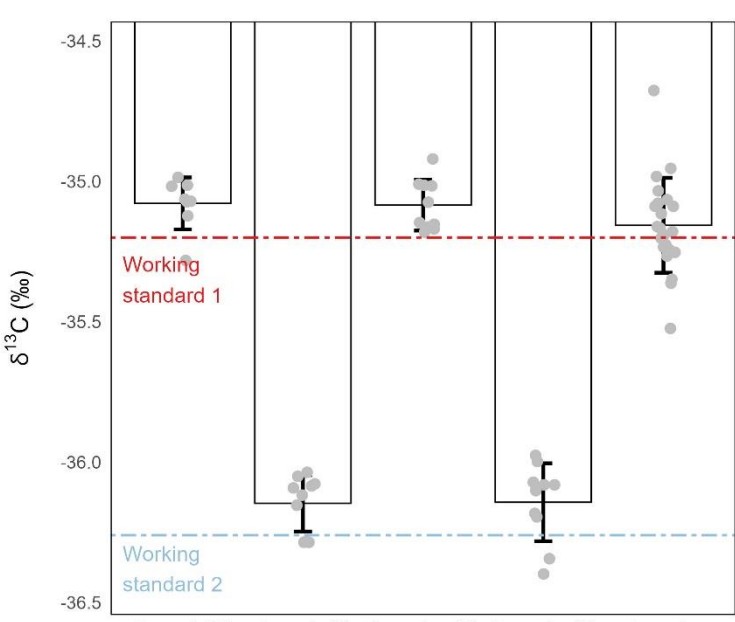

**Figure 7: Effect of storage duration on $\delta^{13}C$ stability. Sample were stored at -80°C for 1- (left), 2- (middle) or 4-weeks (right). $\delta^{13}C$ values (‰ vs VPDB) was measured on 1 mL working standard. The red and light blue dashed lines represent the two expected isotopic value of the working standard's (respectively -35.20 ‰ and -36.26 ‰). Grey circles represent individual replicate**
**measurements. Bars show the mean ± standard deviation for each condition. Precision was ± 0.10 ‰ (n=8) and ± 0.09 ‰ (n=10) for sample stored 1-week (respectively sub samples (1) and (2)), ± 0.13 ‰ (n=10) and ± 0.09 ‰ (n=10) for sample stored 2-weeks (respectively sub samples (1) and (2)) and ± 0.17 ‰ (n=22) for sample stored 4-weeks.**

The results showed that after 1 week of storage, precision was at or below ± 0.1 ‰ for both sample sets (± 0.10 ‰ and ± 0.09
‰; Fig. 7). However, after 2 weeks of storage, precision became more variable, with one sample set exhibiting a precision of ± 0.09 ‰ and the other reaching ± 0.13 ‰ across the two working standard sets (Fig 7). By 4 weeks, precision was no longer acceptable (± 0.17 ‰), indicating a significant evolution of the isotopic signal (Fig 7).

## 4 Discussion

We developed an analytical workflow for measuring the carbon isotopic composition ($\delta^{13}C$) of $CO_2$ in small atmospheric air
samples (1 mL) with high precision (± 0.1 ‰), from sample collection and storage to isotope ratio mass spectrometry analysis. The workflow combines custom vial preparation, optimised storage conditions, and continuous-flow IRMS measure using a GasBench II system, a ConFlo IV and a cryogenic trap. This approach allows for minimal sample disturbance and excellent



precision ($\pm$ 0.1 ‰), while requiring only a fraction of the air volume typically used in $\delta^{13}C$ analyses. This method was designed to meet the specific constraints of experimental systems with limited headspace and sampling flexibility, such as growth chambers, where repeated and minimally invasive gas sampling is essential.

Our results demonstrate that our method enables, for the first time, high-precision $\delta^{13}C$ analysis from as little as 1 mL of ambient atmospheric air, achieving a precision of $\pm$ 0.1 ‰. It opens new avenues for high-resolution, minimally invasive monitoring in experimental setups where sample volume is a limiting factor.

## 4.1 Methodological innovations

A central advancement of our analytical workflow lies in the integration of several critical steps that, individually, may not suffice to ensure analytical precision but, when combined, result in good performance. First, flushing vials with $CO_2$-free air prior to conditioning significantly improved precision of $\delta^{13}C$ analysis. This step appears to remove residual $CO_2$ that may otherwise fractionate or mix with the sample gas, introducing variability. While the importance of vial preconditioning has been acknowledged (Hardie et al., 2010), our data explicitly quantify its effect on sub-millilitre air samples, showing a reduction in standard deviation from $\pm$ 0.54 ‰ to $\pm$ 0.09 ‰.

Second, we evaluated the role of sealing strategy and material integrity. Despite initial assumptions, the use of double septa did not confer measurable benefits over a single septum. However, the application of Terostat® on both and at the bottom of the cap significantly reduced variability during short-term storage, suggesting that leakage and microdiffusion through the septum are non-negligible, especially in small-volume contexts. This finding aligns with earlier reports on gas exchange through septa (Kuehfuss et al., 2014), but provides a quantitative benchmark for small sample volumes.

## 4.2 The critical role of temperature in preserving isotopic integrity

Our results unequivocally demonstrate that ultra-low storage temperature is essential to maintain the isotopic integrity of trace-level $CO_2$ samples (Fig. 6). While dual-sealing improved short-term stability at room temperature (Fig. 5), only storage at -80°C preserved precision within the desired $\pm$ 0.1 ‰ threshold (Fig. 6). This ultra-low temperature storage is crucial for preventing any alteration of the vial contents, particularly the loss of $CO_2$ or isotopic drift, which could compromise precision of subsequent measurements.

This temperature sensitivity reflects the susceptibility of $CO_2$ to interact with vial surfaces and sealing materials, as well as the thermodynamics of gas diffusion. At -80°C, kinetic processes that may alter isotopic signatures (e.g., adsorption-desorption, gas permeation) are lowered. This insight has immediate implications for field and lab workflows, allowing greater flexibility in sampling and batch processing without compromising data quality.

This method of storage is particularly advantageous for long-term analyses, where maintaining the stability of the isotopic signal is essential. Under these conditions, the vials can be stored for extended periods (Fig. 7), ensuring the integrity of the samples.





### 4.3 Limits of storage time and implications for experimental design

Although ultra-low temperatures effectively stabilise isotopic composition (Fig. 6), our data show that this protection is not indefinite. After one week at -80°C, sample integrity remains high, but beyond two weeks, a gradual deterioration in precision is observed (Fig. 7). After four weeks, the isotopic signal becomes significantly altered, limiting the window for reliable analysis (Fig. 7).

This temporal limit suggests that while our method allows short-term storage of small atmospheric volume samples, it does
not fully substitute for real-time or near-term analysis. Therefore, high-frequency sampling campaigns in remote or difficult-to-access environments must account for the logistical need to analyse samples within 7–10 days post-collection, or risk losing analytical resolution.

Interestingly, the observed shifts in $\delta^{13}C$ were bidirectional — with both increases and decreases relative to the target values — suggesting that the deviations were not due to systematic drift, but potentially due to isotopic fractionation via $CO_2$
adsorption/desorption or diffusion-driven gas exchange through imperfect seals. Even at -80°C, minor temperature gradients or inconsistent sealing among vials could introduce variability in $CO_2$ solubility or pressure, potentially leading to partial pressure differences (e.g., off-gassing or condensation inside vials) and variability in headspace composition. While such bidirectional shifts point away from a single dominant mechanism, in many cases we observed a trend toward $^{13}C$ enrichment. This pattern may indicate external contamination and/or preferential leakage of $^{12}C$-enriched $CO_2$ from the vials, further
supporting the role of seal integrity and isotope-selective gas exchange in driving these anomalies.

These effects, while negligible at larger volumes, may become significant at the millilitre scale due to the small absolute quantity of $CO_2$. This highlights the importance of limiting storage duration to no more than 1–2 weeks under these conditions for high-precision $\delta^{13}C$ analysis.

### 4.4 Broader applications and future directions

The method developed here significantly broadens the scope of stable isotope applications in ecological and environmental research, especially in settings where sample volume or accessibility is limited. In systems such as rhizosphere or microscale soil flux chambers—where $CO_2$ concentrations often exceed ambient levels—the method can be further miniaturized, allowing for precise isotopic analysis from sample volumes as small as 50 µL. This flexibility opens promising perspectives for in situ investigations of carbon dynamics in highly confined or sensitive environments.

Furthermore, this approach can be readily adapted for other trace gases or stable isotopic systems (e.g., $\delta^{18}O$ of $CO_2$), pending appropriate calibration and validation. Future work may focus on improving the long-term stability of samples, potentially through alternative sealing materials, as well as automating sample preparation.

To enable off-site sampling in locations distant from the IRMS facility, we also tested a dedicated protocol for transporting and handling exetainers under ultra-low-temperature conditions. Flushed, sealed and conditioned vials were placed in zip-
locks bags filled with $CO_2$-free air and stored in an isothermal box containing dry ice or cold packs. This configuration allowed



safe transport to the sampling location. Following sample collection, the top of each vial was sealed again with fresh Terostat®
and placed in the same bags, filled in the worst case few hours later with $CO_2$-fre air before to be returned to -80 °C storage
until analysis. This test confirmed that vials conditioned up to 24 hours prior to sampling and stored at -80 °C maintained full
isotopic integrity, enabling flexible logistics without compromising data quality.

Finally, combining this sampling method with miniaturised volumes with real-time $CO_2$ flux measurements (e.g., via infrared
gas analysers) could facilitate continuous isotopic monitoring in situ, thus providing powerful new tools for constraining
carbon budgets and tracing ecosystem processes under dynamic conditions.

**Conclusion**

This study presents a novel methodological workflow for carbon stable isotope ($\delta^{13}$C) analysis of atmospheric $CO_2$ from gas
samples as small as 1 mL, achieving an analytical precision of ± 0.1 ‰. By combining rigorous vial preparation, a dual-sealing
strategy using Terostat®, storage at -80°C, and an optimised continuous-flow IRMS setup with cryogenic trapping, our method
addresses key challenges related to sample volume, isotopic drift, and storage stability.

Its high precision and low sampling footprint enable high-resolution spatio-temporal tracking of $CO_2$, facilitating the
investigation of dynamic processes such as autotrophic/heterotrophic respiration, photosynthesis, and carbon fluxes driven
environmental factors like temperature, humidity, or vegetation composition (Bowling et al., 2008; Keeling, 1958; Pataki et
al., 2003).

Beyond controlled environments, this method also opens new perspectives for field sampling where volume or sampling
frequency are limiting factors. Potential applications include:

- Field campaigns requiring delayed sample analysis (e.g., high-altitude sites, tropical forests, frozen soils),
- Long-term monitoring networks where automated small-volume sampling could reduce costs and environmental
disturbance (Ciais et al., 2014).

Moreover, the demonstrated ability to store samples at -80°C for up to a week without significant isotopic alteration greatly
enhances logistical flexibility, particularly for international collaborations or multi-site field campaigns.

In summary, this methodological workflow provides a robust and precise tool for $\delta^{13}$C analysis of atmospheric $CO_2$, improving
the spatial and temporal resolution of carbon cycle studies across a wide range of experimental and natural contexts.

**Data availability**

**Authors contributions**

JS and CP designed the project. JS, CP, and MLT carried out experiments. JS and CP analysed the data from IRMS and AM
performed statistical analyses. JS, CP, MLT and AM wrote the manuscript.



**Competing interests**

The authors declare that they have no conflict of interest.

**Acknowledgements**

This study benefited from the CNRS resources allocated to the French ECOTRONS Research Infrastructure, from the Occitanie Region and FEDER investments as well as from the state allocation 'Investissement d'Avenir' AnaEEFrance ANR-11-INBS-0001.



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
