# Peer review of "High-precision $\delta^{13}$ C-CO2 analysis from 1 mL of ambient atmospheric air via continuous flow IRMS: from sampling to storage to analysis."

_EGUsphere, 2025_

## Author Comment (AC1)

**Response to RC1**

We sincerely thank Reviewer 1 for the very detailed and constructive review. Below we provide a point-by-point response to all comments. Line numbers refer to the revised manuscript.

Sauze et al., (2025) developed a method for the preparation and handling of small air sampling vessels with a volume of 1 mL STP for the analysis of stable carbon isotopes in $CO_2$. The authors describe a series of experiments that lead to a proposed protocol for the preparation and handling of the air sample containers. They find the preparation steps are critical to achieve their best precision. The authors suggest this method opens a door to new analyses of stable carbon isotopes in $CO_2$ from environments where the amount of available sample volume is critically limited, such as rhizosphere and chamber studies. Those applications hold the potential for important improvements of our understanding of carbon cycle processes, which is of great importance to atmospheric science and biogeochemistry, and the future of life on Earth as we know it in general.

I share the view that sampling volumes that small for high precision analyses is technically challenging. The topic of the manuscript is well suited for publication in AMT. However, I'd suggest significant rewriting, including further analysis existing data, and potentially making additional measurements. Some of the results do not seem to be of sufficient quality. The manuscript lacks fundamental basics on conventions and guidelines in isotope research. I will list a few examples and refrain from commenting on details.

We thank the reviewer for recognising the technical challenge of achieving $\delta^{13}C$–$CO_2$ analysis from only 1 mL of air. We acknowledge that the manuscript required clarification and restructuring to improve transparency, understanding, and reproducibility in other laboratories. In response, we have added methodological details where needed.

Given that several comments from the three reviewers highlighted similar concerns, we have also substantially revised the manuscript to more clearly emphasise the core motivation and originality of our approach: developing a simple, low-cost, and rapid method based on standard, commercially available equipment, specifically tailored for analysing very small air volumes. These points were not sufficiently explicit in the original version, and the revised manuscript now consistently underlines this rationale throughout.

**Accuracy**
The elephant in the room seems to be the lack of accuracy in the presented experiments. The authors focus entirely on "precision" as a quality objective (I can't remember seeing a definition of precision, but assume standard deviation of a set of experiment results?) Almost all figures, including all of the results underpinning the finally suggested protocol (Figure 7) show significant offsets between target values and the achieved average values of a series of experiments. To me, the lack of accuracy suggests that something is not quite right with the method, and a critical experimental process not sufficiently controlled. The manuscript does not seek to explore the causes for the inaccuracy. I may be convinced otherwise, but the lack of technical details leaves a lot of room for speculation on the lack of accuracy. I suggest the accuracy problem to be fully explored, ideally with additional measurements, and the details to be provided.

We thank the reviewer for this important comment. In the revised manuscript, we now explicitly define the term precision (line 83) and clarify why we focus on this parameter. While repeatability is classically

defined under strictly identical analytical conditions (Belouafa et al., 2017; Squara et al., 2020), we use precision because, although the analytical setup and protocol were constant, multiple operators contributed across analytical sequences. Precision therefore more accurately reflects the conditions under which the measurements were performed.

Regarding accuracy, we agree that the apparent offsets between target and measured values merit clarification. In the initial version of the manuscript, the uncertainties associated with our secondary standards were not explicitly shown. After including these uncertainties in figures 3-7 of the revised version, the apparent offsets are substantially reduced, indicating that the accuracy of the method is better that initially perceived. We additionally performed statistical tests comparing the certified reference value (treated as fixed secondary working standards) to the mean $\delta^{13}$C-CO$_2$ values obtained for each batch and the outcomes of these tests are now discuss in the corresponding sections. Finally, we provide an overall interpretation of how storage temperature significantly improved accuracy, while septum type or dual sealing did not yield any convincing benefits in this regard.

While there is likely room for further optimisation to improve both accuracy and precision simultaneously, we emphasise that, within the constraints of the method (small sample volume, minimal preparation, routine equipment), the achieved performance meets the requirements for studies in ecology and ecosystem science research, which we now better contextualise (section 4.4). This clarification, together with the quantitative assessment of standard uncertainties and the statistical comparison described above, has been added to the revised manuscript.

**Protocol and gases used to prepare sample vials**
The authors report a difference when flushing the vials for 8 s with CO2-free air. Unfortunately, the flow rate of that flushing is not stated, which should be reported to understand the protocol. Afterwards, the protocol includes four cycles of evacuation to 0.1 bar, followed by filling with pure N2. The authors find a substantial reduction in variability of d13C-CO2 values when the initial flush with CO2-free air is included in the protocol (Figure 2). This suggests to me that the residual CO2 is not sufficiently removed by the four evacuation-flush cycles alone. CO2-free air and pure N2 should lead to the same result, if both have sufficiently low CO2 blank, and if both gases were used to quantitatively remove residual CO2. Have the authors tested the effect when pure N2 or CO2-free air are used interchangeably for flushing or evacuation/fill cycles under the same conditions? Demonstrating the effectiveness of the sample vessel preparation with increasing initial flush flow rate and/or flush time, as well as numbers of evacuation-fill cycles would be useful to determine a protocol that leads to accurate and precise values.

We thank the reviewer for this valuable comment. We agree and included the flushing flow rate in the revised manuscript (line 212). Each 5.9 mL vial was flushed with dry CO$_2$-free air at 13.5 L min$^{-1}$, corresponding to ~40 vial-volume renewals per second. This very high flow ensures rapid and complete replacement of the internal atmosphere. For comparison, Steur et al. (2023) flushed 102 × 1 L flasks for 30 min at 3 L min$^{-1}$, i.e. about 90 volume renewals over 30 min, more than two orders of magnitude lower than in our setup. Flushing alone, however, is not sufficient to fully condition the vials. The subsequent four N$_2$ evacuation–refill cycles ensure near-total CO$_2$ removal and leave the vials filled with pure N$_2$, which minimise nitrous oxide formation in the IRMS ion source.

We did not test flushing with N$_2$ instead of CO$_2$-free air, as the presence of O$_2$ during the refill phase could favour nitrous oxide formation. Tests with He and Ar were also not satisfactory due to diffusion or practicality issues and cost. Blank tests performed under higher sensitive peak-detection settings

(minimum peak height = 5 mV; start/end thresholds = 1 mV s⁻¹) revealed small peaks of 10–15 mV in roughly one-third of the blanks (i.e. no peak in the other ones under those settings). However, this background remains well below the $CO_2$ load of actual samples and contributes negligible variability relative to our ± 0.1 ‰ precision target. These results are now reported in the revised manuscript (section 2.1).

**Standard practice in atmospheric science**
It should be noted that preparing vessels for atmospheric sampling and isotope analysis by evacuating, flushing and even thermal treatment is absolute standard practice. A comprehensive protocol for glass flasks was previously published (Steur et al., 2023, DOI:10.1080/10256016.2023.2234594). Successful protocols work based on a large number of gas exchanges within the vessel to eliminate remains from previous samples (memory) as well as dealing with the small but significant amount of surface water on the internal surfaces.

We thank the reviewer for this remark and agree that evacuation and flushing procedures are well established in isotope analyses, as outlined in Steur et al. (2023). Our protocol follows the same principles for moisture removal and leak prevention but is adapted for a different context: the preparation of 1 mL atmospheric samples in septum vials, where aliquot-based approaches used for large flasks (≥ 1 L) are not applicable. We have clarified this distinction and the motivation for a simple, rapid, low-cost and standard material-based protocol in the revised manuscript (line 74).

**Use of d18O-CO2 as indicator of analytical performance**
The protocol described by Steur et al., 2023 is especially relevant for the analysis of oxygen isotopes in CO2, which are not considered in this manuscript. d18O-CO2 can be a very useful indicator for analytical problems. Therefore, I wonder if the d18O-CO2 data could help to identify the cause of the inaccuracy? For the purpose of method refinement, the precision (standard deviation) of d18O-CO2 from different experiments might be useful.

We thank the reviewer for this helpful suggestion. We agree that $\delta^{18}O–CO_2$ can serve as a sensitive diagnostic parameter for detecting analytical issues such as leakage, fractionation or incomplete equilibration, and that its inclusion could further strengthen the evaluation of method performance. In the present study, however, the reference $CO_2$ used for calibration was not assigned a certified $\delta^{18}O$ value, which prevented reliable interpretation of $\delta^{18}O–CO_2$ results.

We added a statement in the section 4.4 acknowledging the potential of $\delta^{18}O–CO_2$ as an additional quality indicator and outlined its integration as a planned improvement in future developments of the method (line 420).

**Control of residual CO2 and possible impact on d13C**
The authors evacuate their sample vessel to 0.1 bar. In other words, around 10 % of the previous gas would still be present in every following preparation cycle. This seems to include the internal volume of the manifold (Figure 1), which is substantial in comparison to the volume of individual sample vials. This could potentially result in different gas compositions across the vials, especially as the manifold is filled with N2 from one side, pushing the gas from the previous filling towards the other side (pump side) of the manifold, where the vials at the pump end may potentially receive larger fractions of the previous gas filling. This contains some degree of speculation on my side, but I am not convinced that four cycles of evacuating to 0.1 bar and filling with N2 are a guarantee for quantitative replacement of the previous gas. It only takes 1 % of atmospheric CO2 with a d13C of around –8 ‰ and 99 % of a

working standard CO2 with a d13C of around –36 ‰ to cause an offset of 0.3 ‰. Even a 1 % blank of a small sample peak might appear relatively small in the blank test the authors performed (line 93). Because of the small sample volume, the presented method is targeting, system banks are very important, which the authors are well aware of. Vessels with septa can easily be evacuated to fractions of a mbar. Why did the authors choose not to evacuate to much lower pressure levels?

We thank the reviewer for spotting this error. The reported pressure was indeed incorrect — vials were evacuated to 0.02 mbar, not 0.1 bar. At this pressure, the residual gas in a 5.9 mL vial is negligible (~0.0001 mL). Repeating the evacuation–refill cycle four times ensures near-complete removal of the previous gas through geometric dilution. We corrected this value and clarified that, under these conditions, residual $CO_2$ is effectively eliminated (line 106).

**Blank test data**
The authors performed blank tests (i.e., line 93), but do not present blank data, instead stating they didn't find a detectable blank signal. I haven't yet seen a system that has virtually no blank. There has always been some blank, and that blank is ideally smaller than the defined limits, above which Isodat/Qtegra automatically identifies a peak. Just because the software does not report the peak, this doesn't necessarily mean there is no blank. Especially when measuring isotopes in small sample quantities, a small blank can have significant impact. I am not totally convinced that the variability in the "no flush" scenario shown in Figure 2 is not resulting from incomplete removal of the previous gas in the vial (memory). The d13C range in the "no flush" experiment is almost 2 ‰. A quick back of the envelope calculation suggests that around 6 % of ambient air CO2 with d13C of –8 ‰ would be needed to shift a CO2 with d13C of –38.7 ‰ by 2 ‰. This amount might well be detectable in the peak sizes of the measurements. It doesn't seem that insufficient memory and blank control are fully explored in the manuscript and the underlying experiments. These information or experiments should be delivered in a manuscript that seeks to establish a new sample vial preparation protocol as the primary objective.

We thank the reviewer for this comment. We are not entirely sure we fully understand the point being raised, but we note that the variability observed in the "no flush" scenario (Figure 3) indeed reflects the effect of incomplete removal of residual gas. This is precisely why the initial dry $CO_2$-free air flush step is essential in our protocol. After this step, followed by the $N_2$ evacuation–refill cycles, blank levels are negligible, and the vials are ready to receive the sample without introducing measurable isotopic bias.

Few words were also added about memory effects (line 171). In the experiment described by Siegwart et al., 2023) highly contrasting samples were analysed in random order—ranging from very high $CO_2$ concentrations (10,000–20,000 ppm) with enriched signatures (up to 3.5 AT%) to atmospheric levels (~420 ppm) with depleted values (~ –10‰). No measurable carry-over was observed, indicating that memory effects are negligible in our setup and require no additional correction. This is now clarified in the revised manuscript.

For peak detection, standard analytical settings typically use a minimum peak height of 50 mV with start/stop thresholds of 20 and 40 mV s$^{-1}$. To more rigorously assess blank integrity, we reprocessed all blank chromatograms using much more stringent criteria, lowering the minimum peak height to 5 mV and the start/stop thresholds to 1 mV s$^{-1}$. Under these conditions, small $CO_2$ peaks (10–15 mV) were detected in about one-third of the blanks (i.e. no peak in the other ones under those settings). This low-level background is well below normal detection limits and contributes only minimal additional variability, remaining negligible compared with the analytical uncertainty of the $\delta^{13}C$ measurements. These detection settings and their rationale are now described in the revised manuscript (line 116).

**Undisclosed modifications to the IRMS instrument**

The authors state their IRMS method includes modifications from Fiebig et al, (2005) to work for ambient CO2 mixing ratios (line 57), and "adapted here for high precision at trace levels of CO2" (line 112). However, there is no reference, no description or proof of what is done differently from Fiebig et al., (2005) and how that improved the analysis. I regard this as essential information. What are "trace-level CO2" in an atmospheric research journal? A fraction of lower tropospheric mole fraction averages?

We thank the reviewer for pointing out the lack of clarity regarding the modifications introduced relative to the method of Fiebig et al. (2005). In the revised manuscript, we now explicitly describe how our protocol differs from the original carbonate-based setup and why these adjustments are required for analysing ambient air samples with very low $CO_2$ content (line 146).

The new text clarifies:
(i) that our method targets atmospheric $CO_2$ at ~420 ppm, corresponding to ~0.2 µg C per analysis, i.e. an order of magnitude less carbon than in Fiebig et al. (2005);
(ii) that our approach analyses $CO_2$ directly in the ambient air matrix rather than in a He-flushed environment;
(iii) the chromatographic modification (PoraPlot Q column operated at 35 °C instead of 70 °C) to ensure better separation at low $CO_2$ levels;

**Insufficient specification of used components**

A large part of the success of the suggested sample vessel preparation protocol seems associated with the use of Terostat. Terostat seems to be a brand name for a range of sealing products and not a unique product. At no place do the authors explain what specific product they use and what it is made of, so it is impossible for a reader to gauge, or even to follow and adopt. Also, the description of how this is applied to top, and bottom could be more detailed, as the authors suggest this is a significant part to achieving high data quality.

We thank the reviewer for this comment. The term "Terostat" was indeed inappropriate and has been replaced with "butyl-rubber compound". We now provide a clear description of the material used: Teroson® RB 81, a malleable self-adhesive butyl-rubber compound with excellent resistance to gas and moisture transfer. In our basic protocol, the compound is first applied over the entire surface of the septum and vial cap (section 2.2). For dual-sealing, it is additionally applied around the lower part of the cap—directly beneath the sealing ridge and near the thread area—to reinforce protection against micro-leakage (section 3.4). These details have been clarified in the revised manuscript (line 137 & 270) to ensure full transparency and reproducibility of the procedure.

**Accurate description of system performance in context of literature**

The authors state that the method is of "high" precision in title and throughout the text (i.e., line 113). However, 0.1 ‰ precision in an air sample of 1 mL STP is not particularly high or novel, i.e., Brand et al, (2016), DOI:10.1002/rcm.7587, achieve 0.04 ‰ for d13C in CO2 on 1 mL or air with GC-IRMS. Schmidt et al, (2011), https://doi.org/10.5194/amt-4-1445-2011, achieve 0.05 ‰ on around 3 mL pre-industrial air sublimated from an ice core sample. These methods have demonstrated accuracy to within their much smaller measurement uncertainty or better. The additional challenge and potentially the source of the additional uncertainty/inaccuracy of the method described by Sauze et al., (2025) may thus be associated with the sample vials, not really with the sample amount.

We thank the reviewer for this remark. We acknowledge that the precision reported by Brand et al. (2016) and Schmidt et al. (2011) is higher than ours, but the analytical conditions are not directly comparable:

- Brand et al. (2016) rely on an aliquot-based approach: a large gas volume (1–5 L) is sampled once, and multiple 1 mL injections are taken from the same reservoir, allowing repeated measurements and statistical averaging. In contrast, our method analyses the entire 1 mL sample in a single injection, meaning no re-measurement or averaging is possible.

- Schmidt et al. (2011) work under much higher signal intensities, reaching ~30 V on m/z 44, whereas our analyses operate at ~2–3 V. The resulting signal-to-noise ratio inherently limits achievable precision. In addition, without accounting for sample preparation, Schmidt et al. analyse 7 samples in 25 h, while our setup allows ~20 samples to be analysed within a standard working day.

Our objective differs fundamentally from these studies and was not (enough) clearly stated in the original manuscript: provide a practical compromise that enables high-precision $\delta^{13}C$–$CO_2$ analysis using basic equipment, with short turnaround time and minimal sample volume, in a way that can be easily reproduced in other laboratories. We clarified this distinction in the revised manuscript (see introduction & discussion).

**Data quality objectives and indicators for instrument performance**
The authors seem to develop their quality control criterion around the precision value of 0.1 ‰. At no point do they provide an explanation why this value is needed for any analytical purpose. The smaller the precision, the better the protocol seems to be the paradigm. Given the precision criterion is the only data quality objective the authors seem to apply, the rationale for the choice of this value should be presented. A good example to question that approach is shown in Figure 6: The values from storage temperatures of –20C show a precision of 0.24 ‰ and are distributed around the target value with good accuracy. In contrast a precision of 0.1 ‰ is found at –80C, but then the values appear inaccurate. Yet, the storage at –80C is preferred because of the better precision, accepting inaccurate results. I'd suggest being very cautious of that rationale. Accuracy and reproducibility are at least as important as precision.

We thank the reviewer for this important and constructive comment. We agree that the choice of a 0.1 ‰ precision target must be justified. Our rationale for using 0.1 ‰ as a benchmark is both practical and methodological. In the atmospheric and ecosystem tracer community, a precision of ~0.1 ‰ on $\delta^{13}C$–$CO_2$ is widely regarded as the upper threshold for high-quality measurements in chamber-based, Keeling plot, and soil–plant exchange studies, where isotopic variations of 0.3–1 ‰ typically reflect biological or mixing processes (Pataki et al., 2003; Tu et al., 2001; Joos et al., 2008; Breecker et al., 2014; Leitner et al., 2023). Achieving 0.1 ‰ precision therefore provides sufficient sensitivity to resolve biologically meaningful changes without imposing excessive analytical complexity or sample volume.

Modern IRMS and field-capable systems, including Gas Bench II configurations, routinely achieve < 0.2 ‰ external precision for $\delta^{13}C$–$CO_2$ from 12 mL vials (manufacturer specification; Giammanco et al., 2017; van Geldern et al., 2014 for field deployments; instrument/metrology development programmes aiming at ~0.1 ‰ uncertainty). Our objective was to reach comparable analytical performance —0.1 ‰ repeatability— but with one order of magnitude less sample gas (1 mL instead of 10–12 mL). This reflects a deliberate balance between analytical resolution, operational simplicity, and experimental constraints such as chamber headspace volume and sampling frequency.

We clarified this rationale in the revised manuscript and explicitly justify the 0.1 ‰ target in the introduction (line 90).

**Composition of applied gases and gas equipment**

Unfortunately, the authors do not disclose the compositions of the applied gases. Besides "ambient" there is no statement on the $CO_2$ mole fractions in any of the applied gases. In a technical manuscript on $CO_2$ isotope analysis, with particular focus on small sample sizes, knowing $CO_2$ mole fractions of the applied gases is essential to understand experimental processes and results. Including data on the composition of used gases is obligatory for such a manuscript. For all experiments, the compositions of the gases and especially the $CO_2$ mole fractions must be stated. The manuscript should be clear on mole fraction scales, impurities, calibration uncertainties, etc., as well as manufacturers and models of pressure regulators on those cylinders.

We agree with the reviewer that the compositions and specifications of all gases used should be explicitly reported. This information has been added in the revised manuscript for the three working standards (WS1, WS2, and WS3; line 177). All working standards were synthetic air mixtures supplied by Air Liquide ("mélange Crystal" France) containing either 380 or 450 ppm $CO_2$, 20 % $O_2$, with the remainder $N_2$. They are calibrated against secondary standards, which in turn are referenced to NOAA primary standards on the RAMCES platform (LSCE, Gif-sur-Yvette, France), with assigned values of $380.0 \pm 0.9$ ppm $CO_2$ and $-8.42 \pm 0.10$ ‰ $\delta^{13}C$–$CO_2$. The "ambient air" cylinder from Air Liquide was verified against the same reference scale and is used with an HBS 200-3-2.5 pressure regulator.

**Isotope conventions**

The manuscript ignores basic isotope conventions. All isotope reference gases used need to be stated with uncertainty and traceability chain (Camin et al., 2025, https://doi.org/10.1002/rcm.10018). This is important best practice, even though this manuscript does not show atmospheric data that a reader can compare to other measurements. The authors may have used a cylinder containing liquid $CO_2$ as one of the reference gases (Figure 3), and possibly for some of the gas mixing etc. It should be stated whether this contains liquid $CO_2$ as well as gaseous $CO_2$, as this may affect the isotopic composition over time or with different use, i.e., when used for mixing. When referring to an isotope ratio, the isotope ($d^{13}C$) is combined with the molecule ($CO_2$) as $d^{13}C$-$CO_2$ or $d^{13}C(CO_2)$ when referred to isotope values, where the "delta" is italicised. Negative isotope values are expressed with a long dash, rather than simple dash (Coplen 2011, https://doi.org/10.1002/rcm.5129)

We thank the reviewer for highlighting this point. In accordance with best practice (Coplen 2011; Camin et al., 2025), and as mentioned before, we now report all reference gases (WS1, 2 & 3) with their full specifications, uncertainties, and traceability (line 177). No liquid $CO_2$ cylinders were used at any point, ensuring that phase-change fractionation cannot occur.

We also corrected all isotope notation: $\delta$ is now italicised, isotope ratios are consistently written as $\delta^{13}C$-$CO_2$, and negative values use the appropriate en-dash. These corrections bring the manuscript fully in line with current isotope-reporting conventions.

**References**

The manuscript includes a lot of old references. There is nothing wrong with old references and credit should be given to original ideas, but in many cases, things have moved on and improved over several decades.

We thank the reviewer for this remark, we have updated the manuscript by adding several recent and relevant references throughout the revised manuscript.

---

## Author Comment (AC2)

**Response to RC2**

We sincerely thank the reviewer for carefully evaluating our manuscript and providing constructive and insightful comments. Line numbers in our responses refer to revised manuscript.

**General Comment**

This manuscript presents methodologies of vial treatment and continuous-flow IRMS measurement for $\delta^{13}C$-$CO_2$. They are fairly developed for ecosystem measurement applications and worth being shared in the isotope measurement community; therefore, this study is well within the scope of the journal. However, I do not recommend publication of the current manuscript until the following issues are addressed. In addition, Referee#1 already provided a long list of constructive comments, which I totally support and express sincere respect to the thorough evaluation.

We thank the reviewer for the feedback and for highlighting the relevance of our work to the isotope measurement community.

Given that not a few papers on continuous-flow $\delta^{13}C$-$CO_2$ measurement in similar principles are already available over the last 20 years (those both cited and not cited in this study), I do not find the method novel, in contrast to what the authors write. They improved sampling and laboratory treatment methodologies for a reduced sample amount, and these careful descriptions are of value. However, from my point of view, many places in the manuscript look exaggerated in particular for novelty and outlook. I strongly recommend toning down with focus on actual application fields only.

We thank the reviewer for this remark. We agree that the analytical principle is not new and have therefore toned-down statements about novelty throughout the manuscript. Following comments from all reviewers, we now clearly emphasise that the contribution of our work lies in its practical application to volume-limited systems (e.g., growth chambers, microcosms, respiration closed systems chambers, etc.) and in providing a simple, low-cost, and rapid method based on standard equipment for reliably analysing very small air volumes. The revised manuscript reflects this clarified and more moderate positioning (lines 17, 74, 340).

One simple way to minimize storage effects is to employ a glass flask with a stopcock valve with a rubber O-ring, which has been long used in the atmospheric monitoring community (e.g., Worthy et al. 2023). The flasks are in many cases much larger (up to litres), but stability of sample air has been ensured for months to a year as the flaks are transported worldwide including Antarctica. In this regard, one might wonder why the authors take riskier septum, despite of availability of the safer method. I surmise there are several reasons such as practice in field deployment, familiarity and cost, but I did not find them well explained in the manuscript.

We thank the reviewer for raising this point. We agree that the systems mentioned—glass flasks with stopcock valves—are highly reliable and well known to ensure long-term isotopic stability. However, due to their large internal volume, they are not compatible with our experimental constraints. We acknowledge that the original manuscript did not clearly explain the rationale behind our choice of septum-sealed vials. Our approach was specifically guided by the need for simplicity, low cost, rapid handling, and the use of standard commercially available materials, while enabling the analysis of very small air volumes (1 mL).

We have now made these motivations and constraints explicit throughout the revised manuscript, including in the introduction, materials and methods, and discussion.

Throughout the manuscript, precision, which the authors consider important, is not consistently evaluated and described. In this type of experiments, I suppose that efforts are made to minimize measurement uncertainty, which includes uncertainties associated with every measurement process such as pre-treatment, storage, and isotope analysis. These processes affect both variability (precision) and systematic offset (accuracy). I believe that the authors would like to ensure minimized systematic offsets with acceptable measurement variability. Description and discussion on the former are incomplete in the current manuscript, thereby it is difficult to consider that the present method was evaluated sufficiently. Regarding the latter, the authors should define "precision" explicitly throughout the manuscript. I would use "repeatability" or "reproducibility" depending on the context.

We thank the reviewer for this remark. In the revised manuscript, we now explicitly define what we mean by precision (line 83). We chose precision because, although the procedure and instrument remain constant, the operator may vary between analytical sequences, making precision the most appropriate term for our workflow (Belouafa et al. 2017; Squara et al., 2020). These definitions and adjustments are now consistently applied throughout the manuscript.

To address accuracy, we (i) added the uncertainties of our secondary standards to Figures 3–7, (ii) performed one-sample t-tests comparing each batch to the certified reference value, and (iii) discuss these results in the relevant sections. These additions show that most apparent offsets were overestimated in the original version and that accuracy is mainly improved by storage temperature, not by septum type or dual sealing. Altogether, the revised manuscript now evaluates both precision and accuracy more completely and transparently.

The authors presumably consider that "precision" of 0.1 per mil is a criterion to evaluate quality of the measurement, but it is not clear why this value is appropriate. If the authors assume ecosystem measurement applications, one would see relatively large variability through a series of samples (e.g., some ‰), therefore relaxed "precision" might be acceptable.

We thank the reviewer for this important comment. We chose a precision target of ±0.1 ‰ for $\delta^{13}C$–$CO_2$ based on both practical and methodological considerations. As mentioned application targeted, typically biologically meaningful processes, exhibits $\delta^{13}C$ variations of 0.3–1 ‰ (Pataki et al., 2003; Tu et al., 2001; Joos et al., 2008; Breecker et al., 2014; Leitner et al., 2023). Achieving ± 0.1 ‰ precision allows these changes to be resolved reliably.

Our objective was also to reach comparable analytical performance with standard IRMS setups (i.e. precision <0.2 ‰ with 10–12 mL samples) (Giammanco et al., 2017; van Geldern et al., 2014), using only 1 mL of air, balancing analytical resolution, operational simplicity, and constraints of small-volume experimental systems.
This rationale and justification for the 0.1 ‰ target have now been clarified in the revised manuscript (line 90).

I do not think that Table 1 is useful for readers, because the laboratory settings could change and some part of information is clear from Figure 3. Alternatively, a schematic of the whole measurement system including a sample vial connection (GasBench), ConFlo, and IRMS might be presented. The schematic along with the current Figure 3 should be presented in section 2.3.

We thank the reviewer for this helpful suggestion. Following this recommendation, we removed Table 1 and added a schematic representation of the complete measurement system adapted from Gas Bench II operating manual (Thermo Fisher Scientific., 2018)—including the sample vial connection to the Gas Bench, the ConFlo interface, and the IRMS (now Fig. 2A)—alongside the chromatographic output (now Fig. 2B previous Fig. 3). Both elements are now presented together in Section 2.3 to provide a clearer and more coherent description of the analytical workflow.

Below my specific concerns are also detailed, but I think the authors might need corrections which span the entire manuscript, as several comments apply to several relevant places.

**Specific Comment**

P1 L10: The "stable" carbon isotopic "ratio", if only $\delta^{13}C$ of $CO_2$ is referred. One might include $^{14}C$ if the term "composition" is preferred. Modified (line 11).

P1 L11: …, yet its extended application remains limited due to analytical and sampling restrictions. Modified (line 12).

P1 L14: "Terostat" here and other first place where it appears, the authors should describe what it is, not the trade name, e.g., sealant/adhesive tape? A brief description has been added (line 137), and the term "Terostat" has been replaced throughout the manuscript with butyl-rubber sealing compound.

P1 L19: "carbon stable" at another place, "stable carbon" was used. Harmonised throughout the manuscript.

P1 L23: "composition" to "ratio". The bracket "($\delta^{13}C$)" should come just after "ratio", i.e., "The stable carbon isotopic ratio ($\delta^{13}C$) of atmospheric $CO_2$…" Corrected (line 26)

P2 L37: It is weird that the authors mention to the analyzer of Picarro as emerging instrument, but no reference paper is cited. The authors later cite Sperlich et al. (2022), but their instrument was not Picarro. As Picarro is not the only availability, these sentences might be reformulated in a more balance way. Sentence reformulated and reference added (line 39).

P2 L40: Picarro could measure sample air with 1/10 atmospheric $CO_2$ concentration (e.g., ~40 ppm), but it would make the precision worse. According to their data sheet, detection limit is not clearly defined, and also in principle it would not make sense. I would think that it is matter of whether the worse precision at low CO2 concentration is acceptable. Moreover, dilution measurement with laser-based instrument would cause different problem; change in matrix could cause measurement offset. The sentence has been revised to more accurately reflect the performance and limitations of laser-based analysers at low $CO_2$ concentrations (line 42).

P2 L45: …during "mixture between atmospheric and ecosystem reservoirs." Corrected (line 48).

P2 L58: At other places, the term "concentration" is used. Is "mixing ratio" used with a different meaning? Corrected, use only "concentration" in the revised manuscript.

P3 L86: "0.5 bar" is this absolute pressure or above ambient? Above ambient, this has been specified in the text (line 107).

P4 Figure 1: I wondered that there may be a close valve at the vial side of the 3-way valve, otherwise the vials cannot be evacuated with N2 or exhaust connected. The 3-way valve (SS-43GXS4) serves three functions: closing the manifold, opening the $N_2$ line, or opening the exhaust, eliminating the need for an additional shut-off valve. This is clarified in the manuscript (lines 106, 108 and 113).

P4 L107: Explain about "Terostat" here, so that readers without prior knowledge about the product can follow. And how was it used at where of the vial? Describe explicitly ("apply" it, as in the manuscript, says almost nothing). Corrected (line 137).

P4 L109: As in my earlier comment, a schematic figure including the whole measurement system might help readers. The figure in the reference (Fiebig et al. 2005) represent only part of the system and the authors should present the system more in detail as they improved from the earlier one. Figure 2 in

Section 2.3 presents a schematic of the complete measurement system, together with a representative chromatogram.

P4 L117: What is the cryofocus unit? Is it a tubing or a capillary of which material and size? A cryogenic trap is a U shaped 1/16 stainless steel tube fixed to a pneumatically operated lifting unit and placed above a 2 litre dewar of liquid nitrogen, these details were added in the revised manuscript (line 143).

P5 L122: "the cryofocus was raised" to "the cryofocus column (*or tubing as appropriate*) was lifted out from liquid nitrogen" Corrected.

P5 L129: This sentence does not describe traceability clearly. The authors should mention to the international reference material (RM) to which the reported values are traceable eventually. It is unclear if the authors determined $\delta^{13}C$ value of the working standard against an RM or if they have a suit of different gases whose $\delta^{13}C$ values were determined against an RM. We have added a detailed description of the calibration hierarchy and traceability in the revised manuscript. Specifically, we now clarify that the three ambient air working standards (WS1–3) used for $\delta^{13}C$–$CO_2$ calibration were calibrated against secondary reference gases, which are themselves traceable to NOAA primary standards on the RAMCES platform (LSCE, France) line 177).

P5 L133: "…, and analytical precision consistently reached ±0.1 ‰ for $\delta^{13}C$" here and other places, it is unclear if "reached" means whether the value is larger than 0.1 ‰ or not. I would avoid ambiguous verbs or adjectives when large or small matters. For instance, here the sentence could be rephrased like: …analytical precision of $\delta^{13}C$ was consistently <0.1 ‰. Note my comment on "precision" and consider it consistently throughout the manuscript. Ambiguous terms such as "reached" have been replaced with precise formulations indicating whether the precision is above or below the target threshold.

P5 Table 1: See my earlier comment. Removed.

P6 L147: In section 2.1, the authors described ($O_2$-free) $N_2$ was used to prevent possible $N_2O$ production at ion source, but here they use air containing both $N_2$ and $O_2$. Reformulated.

P6 L155: It is unclear if the $\delta^{13}C$ value is a nominal one from the gas company (Air Liquid) or that determined by the authors' laboratory so as to be traceable to an RM. This is important because it would be no wonder if the values have independent traceabilities. We have clarified in Section 2.3 that the three working standards (WS1–3) used for $\delta^{13}C$–$CO_2$ calibration are secondary standards prepared in-house and calibrated against reference gases that are themselves traceable to NOAA primary standards (RAMCES platform, LSCE, France) (line 177).

P6 L158: See my earlier comment. If the term "precision" refers to standard deviation, define so at early place of the manuscript. Now defined at the end of the introduction (line 84).

P7 L169: This section and Figure 3 could be merged into section 2.3. We have merged the figures as suggested: the content of the original Figure 3 is now incorporated as Figure 2A and 2B within Section 2.3. However, we have retained Section 3.2 as a separate section because it presents a result that is meaningful only after demonstrating the effect of flushing with dry $CO_2$-free air.

P8 L186: "50 μL" this amount depends on the $CO_2$ concentration of sample air. How much concentration corresponds to the sample amount 50 μL? The $CO_2$ concentration in the sample corresponding to the 50 μL aliquot ranged between 10,000 and 20,000 ppm.

P8 L194: Same comment as P5 L133. Does "exceed" mean smaller or larger than 0.1 ‰? Corrected (line 252).

P9 L210: "reduce" delete "s"; the risk of gas leakage, diffusion "and associated" isotopic drift. Corrected (line 271).

P10 L233: Same comment as P6 L155. See details added on line 177.

P11 L238: "…improved measurement precision" it seems to me more important that the low temperature storage resulted in the values in agreement to the nominal value, than the magnitudes of the error bars. In the revised manuscript, we now emphasise accuracy more clearly. Each figure includes the nominal value of the working standard together with its ± 0.1 ‰ uncertainty range, enabling direct visual

comparison. We also added one-sample statistical tests comparing measured means to the certified value. These analyses show that low-temperature storage does not merely improve precision, but—more importantly—produces $\delta^{13}$C-$CO_2$ values statistically indistinguishable from the nominal standard.

P11 L257: Same comment as P6 L155. See details added on line 177.

P12 L274: "composition" to "ratio" Corrected.

P13 L297–L301: For integrity, the authors should discuss total uncertainty, not only "precision." In the revised version, the paragraph now explicitly addresses total uncertainty by discussing both precision and accuracy relative to the working standard, and by discussing the physical mechanisms (diffusion, permeation, adsorption–desorption) that influence systematic offsets during storage (line 377).

P13 L302–305: Appropriate reference should be mentioned. Otherwise, readers cannot understand whether the description is an established knowledge or the authors' speculation. At least I do not find this discussion convincing with the current style. Ghiara et al. (2025) provide experimental evidence that $CO_2$ diffusion and permeation through elastomeric materials decrease sharply at cryogenic temperatures, supporting our interpretation of reduced leakage and isotopic drift at $-80\,°C$. Similarly, Sreenath and Sam (2023) demonstrate that low-temperature conditions suppress gas transport in polymer–membrane systems used for $CO_2$ capture. These two studies have now been cited to the relevant section (Section 4.2).

P14 L320: As the authors explain at P14 L323, I agree that diffusion is plausibly the dominant process that caused the result with $\delta^{13}$C offset and larger variability. The authors might come up with adsorption or desorption, but I could not find any signal that support these processes occurring. This paragraph could be reformulated. We have reformulated the paragraph to clarify that while diffusion is likely the dominant process behind the $\delta^{13}$C-$CO_2$ offset and variability, minor contributions from $CO_2$ adsorption/desorption on vial surfaces or elastomeric seals cannot be excluded, especially in small-volume samples. We support this point with recent literature (Aoki et al., 2022; Schukraft et al., 2022; Ghiara et al., 2025; Sreenath and Sam, 2023). See sections 4.2 and 4.3.

P14 Section 4.4: I think discussion of this section is too general and thereby reads exaggerated. For instance, isotope equilibrium with water vapor during storage largely matters in $\delta^{18}$O of $CO_2$ measurement, thereby it would go much less smoothly. The authors additionally explain prior treatment of vials, but storage over days, weeks and months are totally different, and I would not be optimistic as written currently. If the authors have a specific application plan deemed feasible only with the result presented in this study, they could focus on it in this section, but otherwise I think it is difficult to keep this section in a style convincing to readers. Modifications have been made to refine the adaptation for $\delta^{18}$O–$CO_2$, limit extrapolation regarding storage, and clearly highlight the applications under testing or in realistic conditions (section 4.4).

P15 Conclusion: The conclusion (as well as abstract) section should be reformulated after all issues are addressed. Done.

P15 L360: I do not think that the exact methods in this study can help reduce cost of long-term monitoring network, because they analyse $\delta^{13}$C of $CO_2$ using sub aliquot of original flask samples (use of extra vials would increase cost). We removed the statement regarding long-term monitoring networks. Instead, we highlight more appropriate applications, such as long-term laboratory incubations and specialised microcosm experiments, as well as inter-laboratory sample exchange, where small-volume $\delta^{13}$C–$CO_2$ sampling provides a clear advantage.

P15 L362: A week is too short for international collaborations. Extra efforts to cool samples would also complicate logistics. The following sentence also reads like overstatement. Removed.

---

## Author Comment (AC3)

We thank the Reviewer for this constructive review. Below we provide a point-by-point response to all comments. Line numbers refer to the revised manuscript.

Sauze et al. (2025) developed an analytical workflow to measure the carbon isotopic composition of $CO_2$ in small atmospheric air samples (1 mL). Their work demonstrates that all the successive steps described in the paper are necessary to achieve a good precision (0.1‰). The ability to measure isotopic composition in such small quantities opens new perspectives in ecological and paleoenvironmental research. For this reason, the study fits well within the scope of the journal.

I find this study interesting, particularly in the way it tests different parameters. However, as already noted by the other two reviewers, several points need to be addressed before publication.

We thank the reviewer for the positive evaluation of our study and for highlighting the relevance of developing an analytical workflow capable of measuring the carbon isotopic composition of $CO_2$ in very small air samples. All comments have been addressed point-by-point in the detailed responses below, and corresponding modifications have been implemented in the revised version of the manuscript.

I also believe that the authors should explain in more detail why achieving a precision of 0.1‰ is novel. This could be related to Berryman's 2011 study, where a precision of 0.3‰ was reported.

We thank the reviewer for raising this point. We agree that the relevance of a 0.1 ‰ precision target need to be clearly justified and we have clarified this rationale in the revised manuscript (line 90).
In atmospheric and ecosystem applications, $\delta^{13}C$–$CO_2$ signals typically vary by only a few tenths of a per mil, and 0.1 ‰ is widely considered the precision required to resolve meaningful biological or ecosystem-level processes.

Current IRMS setups (including Gas Bench II systems) generally deliver precision <0.2‰ using 10–12 mL of air. Our aim was to reach comparable performance with 1 mL of sample—an order-of-magnitude reduction—while maintaining analytical robustness and compatibility with small chamber volumes and high sampling frequency.

**Quality of the figures**

Although this study focuses on optimizing precision in $\delta^{13}C$ measurements, it is difficult—based solely on the figures—to assess the difference or similarity between the measured precision and the accepted values reported in the literature.

For example, in all figures the mean value is shown in red, but what are the corresponding precisions and accepted standard deviations? How far are the measured values from the reference ones?

We thank the reviewer for this comment. In the revised manuscript, figures 3-7 have been updated to display the reference $\delta^{13}C$ value (red dashed line) and its associated uncertainty range (± 0.1‰) as a shaded band. This allows direct visual evaluation of how each treatment, condition, or protocol compares with both the reference value and its accepted precision.

**Need for more quantitative information**

Since this study is based on the analysis of very small amounts of $CO_2$, more quantitative detail and precision are needed. For example:

- L93: "A control set of vials (i.e., blanks) was analyzed and showed no detectable $CO_2$ signal."
- L249: "These blank vials showed detectable $CO_2$ signals, confirming the integrity of the sealing protocol and the absence of background contamination during storage under ultra-low-temperature conditions."

The authors mention possible contamination, but without quantifying the blanks and background, how can one assess the effect on the results? If the $\delta^{13}C$ of atmospheric $CO_2$ is around –8‰, could this explain the higher $\delta^{13}C$ values measured compared to the reference value of the working standard? A possible mixing process could be discussed.

We thank the reviewer for this constructive comment. To quantitatively assess blank integrity, we reprocessed all blank chromatograms using higher sensitive detection settings (minimum peak height 5 mV; start/stop thresholds 1 mV s$^{-1}$), well below standard analytical thresholds used here. Under these conditions, small $CO_2$ peaks (10–15 mV) were detected in approximately one-third of the blanks (line 116).

We now also discuss the theoretical possibility of mixing with atmospheric $CO_2$ ($\delta^{13}C \approx -8$‰). Given that blanks remain below detection limits, any such contribution would represent a negligible fraction of the $CO_2$ in a sample and would fall well within analytical uncertainty (line 361).

**Accuracy**

Although this study focuses on improving $\delta^{13}C$ precision, it does not address accuracy. There are often systematic offsets between the mean measured values and the reference values, but these are not discussed. For example, why does the single-septum test (l. 189, 3.3) show an offset from the reference value that is not observed in the other tests?

We thank the reviewer for this comment. We agree that the apparent discrepancies between target and measured values required further clarification. Accuracy is now explicitly addressed. In the revised figures, we display the working-standard reference value together with its ± 0.1‰ uncertainty envelope, enabling direct comparison. The apparent offsets are considerably reduced, showing that the method is more accurate than initially assumed.

We also performed statistical tests comparing the certified reference value (treated as fixed) with the mean $\delta^{13}C$–$CO_2$ values obtained for each batch, and the results of these tests are now discussed in the relevant sections.

Finally, we provide an overall interpretation indicating that storage temperature had a clear positive effect on accuracy, whereas septum type and dual sealing did not yield any convincing improvements (section 4.2).

Even if measurements were performed using a continuous-flow setup, it would be useful to test different IRMS settings. What were the instrumental parameters and operating conditions used during the study? Providing this information could help other researchers reproduce the method.

The analytical setup for $\delta^{13}C$–$CO_2$ is now shown in Figure 2A. All operating details—including carrier gas flow, GC temperature, peak detection thresholds, number of reference injections, autosampler settings, and others—are described in sections 2.1 and 2.2 of the revised manuscript.

Additional aspects that could be discussed include:
1. the impact of instrumental drift during analysis,
2. the memory effect, and
3. the background signal

Data accuracy is strongly linked to background levels, with increasing background often leading to less accurate and less precise results. What are the quantitative values associated with these three parameters, and how were the data corrected for them?

(1) Instrumental drift was monitored through systematic injections of the working standard at regular intervals during each run (at both the beginning and end of each daily sequence). Drift remained below 0.05 ‰ over the longest sequences (4–5 h), which is well within the analytical uncertainty. No time-dependent trend was detected, and this information is now reported in the revised manuscript (line 185).

(2) Memory effects were assessed during the experiment described by Siegwart et al. (2023). Highly contrasting samples were analysed in a random order, resulting in successive injections of $CO_2$ with very high concentrations (10,000–20,000 ppm) and highly enriched isotopic signatures (up to 3.5 AT%), followed by samples with atmospheric $CO_2$ concentrations (~420 ppm) and depleted signatures (~ – 10‰). No memory effect was detected under these conditions, indicating that carry-over is negligible and does not require additional correction or mitigation in our analytical setup. This is now specified in the revised manuscript (line 171).

(3) It is not entirely clear what is expected regarding the background signal. Instrumental background is already corrected by the software through the blanking procedure (line 167). Background originating from the vials is now discussed in the revised manuscript: under highly sensitive detection settings, small $CO_2$ background peaks (10–15 mV) were detected in approximately one-third of the blanks. These correspond to very small amounts of $CO_2$ and remain well below standard analytical detection thresholds (50 mV) (line 361).

Finally, the authors mention diffusion as a plausible cause of some effects. It would be useful to provide references or a brief explanation to support this hypothesis

Ghiara et al. (2025) provide experimental evidence that $CO_2$ diffusion and permeation through elastomeric materials decline sharply at cryogenic temperatures, supporting our interpretation that leakage and isotopic drift are minimized at –80 °C. Likewise, Sreenath and Sam (2023) show that low temperatures substantially suppress gas transport in polymer–membrane systems designed for $CO_2$ capture. These findings are now incorporated into the discussion of diffusion processes in Section 4.2.